# Amelioration Effect of *Lactobacillus kefiranofaciens* ZW3 on Ovalbumin-Induced Allergic Symptoms in BALB/c Mice

**DOI:** 10.3390/foods14010016

**Published:** 2024-12-25

**Authors:** Hanxue Xu, Xiaowei Duan, Yanping Wang, Weitao Geng

**Affiliations:** College of Food Science and Engineering, Tianjin University of Science & Technology, Tianjin 300457, China; xhx312552709@163.com (H.X.); 15513994066@163.com (X.D.); ypwang@tust.edu.cn (Y.W.)

**Keywords:** *Lactobacillus kefiranofaciens*, allergic symptoms, Th1/Th2 balance, intestinal flora

## Abstract

Previous studies have shown that supplementation with specific probiotics can be used to alleviate allergy symptoms. The purpose of this study was to evaluate the anti-allergic effects of *Lactobacillus kefiranofaciens* ZW3 (ZW3) in ovalbumin (OVA)-induced allergic mice. The mice were divided into six groups: the food allergy group, positive group (*Lactobacillus rhamnosus* GG), low-dose ZW3 group, middle-dose ZW3 group, high-dose ZW3 group, and the control group involving healthy mice. BALB/c mice were intraperitoneally injected with OVA/complete Freund’s adjuvant (CFA) for allergy sensitization. Probiotics were administered orally once every two days in the probiotic-treated groups. The allergic score, serum OVA-sIgE, body mass, thymus, and spleen indexes were detected on day 22, and the relative mRNA expression of inflammatory cytokines was detected via RT-qPCR. The results suggest that the body weight and thymus index returned to normal levels; allergy scores, serum OVA-sIgE, IL-4, IL-5, and IL-10 expression decreased; and IFN-γ and IL-2 increased significantly in the ZW3 group compared with the allergy group. Furthermore, ZW3 decreased *Muribaculaceae* and *Ruminococcaceae* abundance and increased *Lachnospiraceae* abundance in the intestinal flora. In summary, ZW3 induced anti-allergic effects by increasing Th1 cytokines and decreasing Th2 cytokines, which can remarkably ameliorate the symptoms of an ovalbumin-induced food allergy.

## 1. Introduction

Allergic diseases present medical researchers with significant obstacles to treating allergies because of their diversity, their complex clinical symptoms, and the long duration of illness [1]. As a kind of allergic disease, food allergy is an adverse immune response to ingesting food antigens. Food allergy can be categorized into IgE-mediated food allergies and non-IgE-mediated food allergies. Among them, IgE-mediated food allergy is a rapid-onset allergic reaction, which usually manifests as symptoms such as abdominal cramps and vomiting or diarrhea within a few minutes to two hours post-exposure, which are even accompanied by fever, hematemesis, hematochezia, or anaphylactic shock in severe cases. This type of allergy is the main form of food allergy [2]. Food allergy has begun to attract the attention of most people, mainly due to the following two reasons: on the one hand, studies have found that the incidence of food allergy is increasing annually; on the other hand, modern food-processing technology has extracted, separated, and synthesized some high purity proteins that may serve as potential allergens [3,4,5]. In food allergy, egg allergy is an IgE-mediated type I hypersensitivity, which is usually caused by ovalbumin, and it is the most common food allergic reaction, with survey data suggesting that egg-induced food allergic reactions account for 50% of all allergic reactions to food in Asia [6,7]. There is no effective treatment for food allergy. Complete avoidance of food antigens is the only effective way to prevent it; however, this is difficult to achieve in clinical practice. At present, the main clinical treatment for food allergy is specific immunotherapy (SIT). Only some patients achieve persistent clinical tolerance despite achieving a high rate of desensitization [8,9].

Emerging evidence suggests that the dysregulation of the intestinal flora, a complex community of microorganisms, may contribute to the development of diseases, such as asthma, celiac disease, and allergies [10]. Several studies have shown that the intestinal flora is closely related to the development of food allergies and the characteristics of the intestinal flora are different for different types of food allergies [11]. *Lachnospiraceae*, *Streptococcaceae*, and *Leuconostocaceae* are enriched in the intestinal flora of patients with an egg allergy [12]. Moriki et al. proved that the relative abundance of *Bifidobacterium*, *Coprococcus catus*, *Monoglobus*, and *Lachnospiraceae* GCA-900066575 in the gut microbiota of children with a milk allergy was reduced compared to healthy children, while *Oscillibacter valericigenes*, *Negativibacillus massiliensis*, and *Ruminococcaceae* were enriched [13]. Researchers performed fecal microbiota transplantation (FMT) studies to further confirm the relationship between the gut flora and food allergy. Noval Rivas et al. suggested that the transplantation of fecal samples from allergic mice to germ-free mice promoted allergic reactions, which proved that intestinal dysbiosis might precede the onset of allergy [14]. In addition, FMT treatment improved the gut microbiota composition and restored the gut microbiota to the pattern of the control group in OVA-sensitized mice. FMT intervention also modulated the Th1/Th2 immune balance and reduced IgE production [15].

Probiotics, as safe and efficient ingredients for modulating the human gut microbiota, have been given more and more attention in anti-allergy studies [16,17,18,19]. Previous studies have shown that compared to standard treatment alone, *Bifidobacterium lactis* Bb-12 or *L. rhamnose* GG, used as an adjunctive therapy, led to an earlier recovery from an allergy [20,21]. The oral administration of *Lactobacillus murinus* restored the deteriorated intestinal flora and alleviated allergic reactions in food-allergic mice [22]. In addition, Hong et al. showed that the heat-inactivated strain of *Lactobacillus kefiranofaciens* M1, isolated from kefir grains, could effectively inhibit the production of IgE and tilt the Th1/Th2 balance toward Th1 predominance to exert anti-allergic activity [23]. In view of the above findings on probiotics improving food allergies, we suggest that microbiota-based therapy has great potential as a strategy to alleviate food allergies.

*Lactobacillus kefiranofaciens* ZW3 (ZW3) was isolated from kefir grains; it has the ability to colonize and modulate gut flora [24]. Previously, we have demonstrated that ZW3 exhibited anti-colitis and anti-depression effects, and an improvement in nonalcoholic fatty liver disease (NAFLD) via regulating intestinal flora in a number of in vitro and in vivo experiments [24,25,26,27,28,29]. In addition, in our previous studies, it was discovered that ZW3 exerted immunomodulatory activities on RAW 264.7 macrophage cells. Therefore, in this study, we explored the ability of ZW3 to exert immunomodulatory and anti-allergy effects in an ovalbumin-induced allergic mice model and further determined whether ZW3 alleviated allergic symptoms by targeting the intestinal flora.

## 2. Materials and Methods

### 2.1. Bacteria Preparation

*Lactobacillus kefiranofaciens* ZW3, which is preserved at the Culture Collection Center of the Institute of Microbiology, Chinese Academy of Sciences (accession number CGMCC2809), was isolated from kefir grains. The ZW3 was suspended in sterile phosphate-buffered saline (PBS) at concentrations of 4.5 × 10^7^ CFU, 3 × 10^8^ CFU, and 1.5 × 10^9^ CFU.

The *Lactobacillus rhamnosus* GG (LGG) used in this study was cultured in MRS broth (Beijing Solarbio Science & Technology Co., Ltd., Beijing, China) at 37 °C for 22 h. The cells were collected using centrifugation at 3000× *g* for 10 min at 4 °C and washed twice in sterile PBS; the cell suspension was adjusted to a concentration of 1 × 10^9^ CFU/mL.

### 2.2. Animals

BALB/c female mice aged 6–8 weeks (15–17 g) (Beijing SBF Biotechnology Company Ltd., Beijing, China) were kept in a specific pathogen-free environment (temperature: 24 ± 1 °C; humidity: 55 ± 5%; 12 h/12 h light/dark cycle). All the experiments conducted in this study were approved by the Institutional Animal Care and Use Committee of Tianjin University of Science and Technology (approval number. TUST2024027).

### 2.3. Sensitization with Ovalbumin

After a 7-day acclimatization period, forty mice were randomly divided into six groups: the control group, food allergy model group, LGG group (1 × 10^9^ CFU/mL), low-dose ZW3 group (4.5 × 10^7^ CFU), middle-dose ZW3 group (3 × 10^8^ CFU), and high-dose ZW3 group (1.5 × 10^9^ CFU). On day 0, the mice in the food allergy model group and the lactic acid bacteria groups were injected intraperitoneally with 0.2 mL of a 1 mg/mL mixed allergen solution (1 mg of ovalbumin (Sigma-Aldrich, St. Louis, MO, USA) mixed with 1 mL of complete Freund’s adjuvant (Sigma-Aldrich, St. Louis, MO, USA)). On days 7, 14, and 21, 0.2 mL of a 5 mg/mL mixed allergen solution was injected intraperitoneally to enhance the immune response. The mice in the control group were injected with PBS at the same time [30,31].

### 2.4. Bacterial Treatment

For the oral administration of bacteria, probiotic strains were suspended in PBS to prepare a bacterial suspension and administered orally to OVA-sensitized mice (200 μL/day) once every 2 days for 4 weeks. At the same time, PBS was administered orally to the mice in the food allergy model group. On day 22, the mice were sacrificed, and the blood, spleen, and thymus were collected for further study.

### 2.5. Allergy Score, Body Mass, and Spleen and Thymus Indexes

On day 22, the allergy scores of each group were recorded. The allergy scores were as follows: 0 for no symptoms; 1 point for scratching the mouth and nose; 2 points for enlarged eyes and mouth, decreased mobility, and increased respiratory rate; 3 points for shortness of breath and blueness of mouth and tail; 4 points for excitement, tremor, and muscle contraction; and 5 points for shock and death.

On day 22, all the mice were sacrificed, body mass was weighed, spleen and thymus were separated, and spleen and thymus indexes were calculated (spleen index (mg/g) = spleen mass/mouse body mass, thymus index (mg/g) = thymus mass/mouse body mass).

### 2.6. Total IgE and OVA-Specific IgE in Serum

On days 1, 8, and 15, blood samples from the tail vein were obtained from both the control group and the food allergy model group and centrifuged at 10,000 rpm/min. The total serum IgE levels were analyzed using ELISA (JONLNBIO, Shanghai, China).

On day 22, eyeball blood was removed, and the serum was centrifuged at 10,000 rpm/min. The serum OVA-sIgE levels were detected using ELISA (JONLNBIO, Shanghai, China).

### 2.7. Real-Time PCR Assay

Total RNA in the spleen tissues was extracted using the TRIzol method, and its concentration was found to be 260 nm and 280 nm. Reverse transcription was used to obtain the cDNA, using the HiFiScript cDNA synthesis kit (CWBIO, Taizhou, Jiangsu, China). The relative mRNA expression of interleukin-2 (IL-12), interferon-γ (IFN-γ), IL-4, IL-5, IL-10, IL-13, and glyceraldehyde 3-phosphate dehydrogenase (GAPDH) was determined using the Bio-Rad CFX96 real-time PCR detection system (Bio-Rad, Hercules, CA, USA). The reaction system consisted of 2 µL of cDNA, 1 µL of each of the forward and reverse primers, 25 µL of 2× UltraSYBR Green PCR Master Mix (BeyoFast™, CWBIO, Taizhou, Jiangsu, China), and 21 µL of sterilized ultrapure water. The primers used in this study are listed in Table 1.

### 2.8. Microbial Profile Analyzed Using 16S rRNA Sequencing

The mouse feces were collected under aseptic conditions and preserved in dry ice. Subsequently, the samples were sent to Biomarker Biotechnology Co., Ltd. (Beijing, China) for microbiological analysis. The diversity analysis of the gut microbiota was performed using the 16S rRNA sequencing method based on the Illumina HiSeq sequencing platform.

### 2.9. Statistical Analysis

The data are presented as mean ± SD (n = 10). SPSS 23.0 software was used to conduct the statistical analysis and normality test of the experimental data. One-way analysis of variance (ANOVA) was used to perform the significance analysis (*p* < 0.05), and the Shapiro–Wilk test was used to perform the normality test.

## 3. Results

### 3.1. Increased IgE in OVA-Sensitized Mice, and ZW3-Mediated Suppression of Serum OVA-sIgE

An intraperitoneal injection of OVA into BALB/c mice was used to establish the food allergy model (Figure 1A). During sensitization, the total serum IgE levels were detected every week in the control and the food allergy model groups using ELISA. An increase in total serum IgE is a typical sign of the successful establishment of an animal food allergy model [32,33]. During the primary immune response to the allergen, the mice in the model group showed ear scratching, cheek scratching, decreased vitality, dyspnea, and hair erection. The synthesis of IgE was significantly increased in the food allergy mice after a second exposure. After the third intraperitoneal injection of OVA, the total IgE in the model group reached its highest level, whereas the total serum IgE in the control group remained unchanged throughout the procedure (Figure 1B). The establishment of the OVA-induced food allergy model was deemed successful, as evidenced by the development of allergic reactions and the increase in the level of OVA-specific IgE in the serum, compared with the control group.

The mice were sacrificed on the 22nd day of the experiment and blood samples were collected to determine the immunological factors. The OVA-specific IgE levels are presented in Figure 1C. Compared with the controls, the OVA-specific IgE levels were notably enhanced in the allergy group (*p* < 0.01). When the bacteria were administered to each group, the OVA-specific IgE levels showed different results. We found that the OVA-specific IgE levels decreased significantly (*p* < 0.01) in the LGG and high-dose ZW3 groups compared with the food allergy model group, whereas no significant differences were observed in the low-dose ZW3 and middle-dose ZW3 groups. This indicated that the oral administration of ZW3 could alleviate an ovalbumin-induced food allergy only when the dosage reached a certain level of bacteria.

### 3.2. The Allergy Symptoms, Body Mass, and Spleen and Thymus Indexes in OVA-Sensitized Mice Orally Administered ZW3

The allergy symptoms in the mice were recorded according to the allergic area. Referring to the allergy scoring criteria, the allergy symptoms of the mice in each group were observed and recorded, and the allergy scores were calculated. Then, the mice were sacrificed. The spleen and thymus were weighed, and the spleen and thymus indexes were calculated. The administration of either LGG or high-dose ZW3 for 21 days attenuated the severe allergy symptoms compared with the control group (*p* < 0.01). However, neither the low-dose ZW3 nor the middle-dose ZW3 had a significant effect on the allergy symptoms (*p* > 0.05). These results indicated that a certain concentration was necessary and that the amelioration of the allergy symptoms associated with ZW3 occurred in a dose-dependent manner (Figure 2A).

The immune organ index is a preliminary indicator used to measure the immunoregulatory capacity of the organism. The spleen and thymus are the main immunoregulatory organs in the body. Therefore, in this study, body weight, the spleen index, and the thymus index were selected to evaluate the effect of ZW3 on the immune organs of the body, and the results are shown in Figure 2B,C. The body mass of the food allergy model group decreased significantly in comparison to the control group (*p* < 0.01), which may be due to the strong allergenicity of OVA, which caused diarrhea in the mice during the allergic process. In addition, the spleen index and thymus index of the food allergy group increased markedly (*p* < 0.01). This may be due to the increased demand for antibody production and the decrease in body mass of the sensitized mice, resulting in enlargement of the spleen and thymus in the allergy model mice. The treatments with high-dose ZW3 and LGG partially or completely suppressed the significant increases in body weight and the spleen and thymus indexes in the model group.

### 3.3. Oral ZW3 Maintains Th1/Th2 Balance in Allergic Mice

Allergies are characterized by a dysregulated immune response to allergens, resulting in an imbalance of Th1/Th2 cytokines. Th1 cytokines are essential for mediating cellular immunity. Th2 cytokines are related to humoral immunity, and the over-secretion of Th2 cytokines leads to allergy symptoms, which in severe cases can lead to anaphylaxis. Cross et al. confirmed that the pathogenesis of food allergies is the result of a preference for the Th2 immune response [34]. Current studies on anti-allergic mechanisms have focused on reducing allergen exposure and maintaining the Th1/Th2 balance. Our research measured Th1 and Th2 spleen cytokines to evaluate the effect of ZW3 on immunity factors. Compared with the food allergy model group, the expression of the Th1 cytokines IL-2 and IFN-γ in the middle-dose ZW3 and high-dose ZW3 groups was notably enhanced (*p* < 0.05). The mRNA expression of the Th2 cytokines IL-4, IL-5, IL-10, and IL-13 decreased significantly in the LGG group (*p* < 0.01); the expression of IL-4 and IL-10 in the high-dose ZW3 group decreased significantly (*p* < 0.01); and the expression of IL-5 in the middle-dose ZW3 group decreased significantly (*p* < 0.01). However, the expression of IL-13 in the ZW3 group showed no significant difference (*p* > 0.05) (Figure 3). These results demonstrated that ZW3 supplementation enhanced the expression of Th1 cytokines; however, the Th2 cytokines were downregulated. Compared with ZW3, LGG could not enhance the expression of the Th1 cytokines, which suggests that the immune regulation of probiotics may be strain-specific.

### 3.4. Effect of Orally Administered ZW3 on Intestinal Flora in OVA-Sensitized Mice

Increasing evidence supports the argument that intestinal microbiota dysbiosis seriously affects food allergy progression [35]. As probiotics improve organismal health by regulating the intestinal flora, we further investigated the effect of ZW3 on the intestinal flora of allergic mice using the 16S rRNA sequencing method. The Shannon and Simpson indexes, which indicate the α-diversity of gut microbiota, increased in the probiotic-treated groups compared with the allergy group; however, there was no significant difference (Figure 4A,B). PCoA analysis showed that the Bray–Curtis distance between the ZW3 group and the control group was reduced after treatment with ZW3 compared with the model group (Figure 4C). These results indicated that there was no significant effect of ZW3 on the diversity of the intestinal flora; however, ZW3 partially reversed the OVA-induced structural changes in the intestinal flora.

At the phylum level, the majority of the bacterial species in each group were Firmicutes, Bacteroidetes, and Actinobacteria. Furthermore, the Firmicutes/Bacteroidetes (F/B) ratio is frequently used to assess the pathological state of the organism, and a lower F/B value is observed in patients with inflammatory and metabolic diseases, among others. Our results suggested that the F/B ratio was increased in the *Lactobacillus*-treated groups; this finding was similar to that of Liu (Figure 4D,E) [36]. As indicated in Figure 4F, at the family level, the relative abundance of SCFA-producing bacteria *Lachnospiraceae* was enhanced after treatment with probiotics [37,38]. SCFAs have been shown to improve intestinal homeostasis and alleviate food allergy symptoms [39]. In addition, the relative abundance of *Muribaculaceae* and *Ruminococcaceae* was upregulated in the OVA group compared with that in the normal group, whereas their relative abundance was downregulated after treatment with ZW3. Canani et al. and Gu et al. previously showed that *Muribaculaceae* and *Ruminococcaceae* were enriched in an allergic group compared with a non-allergic group, which was consistent with our results [40,41].

## 4. Discussion

Allergies are often accompanied by a reduced quality of life and can be life-threatening in severe cases. In the past decades, the prevalence of allergies has steadily increased. The treatment of allergies includes avoiding allergens and administering medications, but there is currently no definitive cure for allergies [42]. Probiotic interventions, as a form of therapy based on modulating gut microbiota, have been shown to be effective in improving allergies [35]. In this study, we demonstrated that ZW3 effectively alleviated allergy symptoms, decreased serum OVA-sIgE levels, improved allergy-induced weight loss, and enhanced spleen and thymus indexes in mice.

The food allergy process consists of two phases: sensitization and elicitation. During the sensitization phase, the allergen enters the organism and makes contact with the antigen-presenting cells (APCs) in the intestinal lamina propria. These APCs present the antigens to T lymphocytes in the gut-associated lymphoid tissue (GALT) through antigen presentation pathways. The T cell receptor (TCR) on the surface of the T cells is activated, which results in the promotion of the differentiation and proliferation of T helper 2 (Th2) cells. The Th2 cells release various Th2 cytokines, which disrupt the Th1/Th2 immune balance and stimulate B cells to produce excessive amounts of IgE antibodies. The excessive IgE antibodies bind to FcεRI receptors on the mast cell and basophil surface; this causes the body to enter the sensitization phase. Then, when the allergen re-enters the body, it specifically binds to the IgE antibodies on the surface of the mast cells and basophils. In response to this stimulation, the mast cells or basophils degranulate, releasing allergic mediators, such as histamine, prostaglandin, and 5-hydroxytryptamine, which induce local or systemic allergic reactions, and the organism enters into the elicitation phase [6,43]. Therefore, restoring the Th1/Th2 immune balance is a critical target for the treatment of allergic diseases. Previous studies confirmed that some specific probiotics, such as *L. casei* Shirota and *L. plantraum* L-137, stimulated macrophages to secrete pro-inflammatory cytokines such as IL-12, which triggered Th1 immunity, to alleviate a casein-induced food allergy. In our results, ZW3 upregulated the expression of the Th1 cytokines IFN-γ and IL-2, downregulated the Th2 cytokines IL-4, IL-5, and IL-10, triggered Th1 immunity, and inhibited Th2 immunity. Among these cytokines, IL-4 promotes a B cell class switch to produce IgE. Excessive IgE antibodies activate basophils and mast cells to synthesize and release a large number of bioactive mediators, which cause local or systemic allergic reactions, particularly chronic allergic diseases. Nawaz M. et al. showed that *Lactobacillus fermentum* NWS29 depressed IL-4 mRNA expression in spleen tissue; in addition, IFN-γ and IL-4 antagonized each other at different levels; these findings are consistent with those of our study [32]. In this study, we also observed that ZW3 could inhibit the expression of IL-10 in the spleen. Hougee et al. also mentioned that probiotics could alleviate ovalbumin-induced food allergy symptoms by reducing the expression of IL-10 mRNA [44].

Increasing evidence suggests that the intestinal flora and its metabolites potentially have the ability to modulate the body’s immune system and improve tolerance to allergens [40]. Li et al. proved that *Lactobacillus reuteri* modulated the intestinal flora, promoted butyrate generation, alleviated airway inflammation, and reduced Th2-associated pro-inflammatory cytokines in asthmatic mice [45]. A probiotic mixture significantly reduced anti-OVA IgE and IgG1 levels, decreased neutrophil recruitment and local IL17 levels, decreased the abundance of *Staphylococcus* and yeast, and improved the composition of the intestinal flora [46]. Furthermore, it has been suggested that *Lactobacillus murinus* performs anti-allergic activity by regulating gut microbiota, the function of intestinal CD11c(+) cells, and the Th1/Th2 balance [22]. The above research suggests that probiotics have positive effects in alleviating the symptoms of many allergic diseases. In our research, ZW3 increased the abundance of *Lachnospiraceae* and decreased *Muribaculaceae* and *Ruminococcaceae.* Interestingly, Xu et al. proved that *Muribaculaceae* might trigger overly vigorous immune responses [47]. *Ruminococcaceae* was reported as a bacteria that was potentially associated with Th2-related responses [48]; *Lachnospiraceae* may inhibit the differentiation of Th2 cells [49]. These results suggest that ZW3 significantly regulated the abundance of bacteria associated with immune responses, thereby alleviating allergies.

Probiotics are considered to be potentially effective substances for maintaining immune homeostasis and preventing or treating allergies, due to their ability to enhance intestinal epithelial barrier function, regulate cytokine balance, and improve intestinal flora. *Lactobacillus kefiranofaciens*, which is from a safe source and has multiple biological activities, has been permitted to be used in functional foods in many countries, including the European Union countries, Japan, Korea, Canada, and China [50]. *L. kefiranofaciens* has been reported to have various health benefits and is used to regulate intestinal flora [24], improve colitis [27], enhance intestinal immunity [51], and ameliorate obesity [26]; it also has anti-bacterial [52] and anti-tumoral properties [53]. However, its anti-allergic activity is rarely reported. Hong et al. previously demonstrated that *L. kefiranofaciens* M1 exerts anti-allergic activity by regulating the Th1/Th2 balance, increasing the proportion of CD4+CD25+ regulatory T cells, and decreasing the activation of CD19+ B cells; their study primarily focused on the effects on cytokines and immune cells [23]. The efficacy of *L. kefiranofaciens* in interacting with the immune system by targeting the gut microbiota suggests that we should also pay attention to the effects of *L. kefiranofaciens* on the modulation of gut flora in an anti-allergy model. Our study showed that ZW3 reduced OVA-sIgE levels and regulated the Th1/Th2 balance, which may be associated with an increasing abundance of *Lachnospiraceae* and a decreasing abundance of *Muribaculaceae* and *Ruminococcaceae* in the gut. This evidence suggests that ZW3 can prevent and treat allergy symptoms by regulating gut microbiota and maintaining immune homeostasis; this will help to provide new insights into the effect of *L. kefiranofaciens* on the improvement of the treatment of food allergies and will provide guidance for the development of *L. kefiranofaciens* probiotic products.

## 5. Conclusions

In summary, our study indicated that *L. kefiranofaciens* ZW3 normalized the immune organ index; increased the expression levels of the Th1 cytokines IFN-γ and IL-2; and decreased the expression levels of the Th2 cytokines IL-4, IL-5, and IL-10, thereby regulating the Th1/Th2 balance and exerting immunoregulatory effects in the ovalbumin-induced food allergy model. Moreover, *L. kefiranofaciens* ZW3 was found to affect the abundance of bacteria associated with immune responses; specifically, it increased *Lachnospiraceae* and decreased *Muribaculaceae* and *Ruminococcaceae,* thereby balancing the intestinal flora. Thus, *L. kefiranofaciens* ZW3 has the potential to be applied in the prevention or treatment of food allergy diseases. Furthermore, the present study demonstrated that *L. kefiranofaciens* exerted anti-allergic effects that might be related to the regulation of intestinal flora; thus, we provided a potential mechanism for its anti-allergic activity in organisms. However, further clinical studies are warranted to confirm the feasibility of *L. kefiranofaciens* in alleviating food allergy symptoms in humans.

## Figures and Tables

**Figure 1 foods-14-00016-f001:**
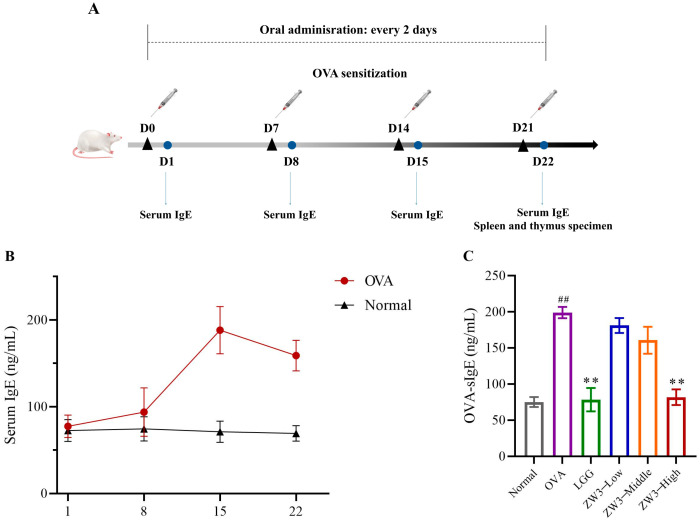
(**A**) Establishment of OVA-sensitized mice model. Serum total IgE levels in control group and OVA group during sensitization (**B**) and serum OVA-sIgE levels in all groups (**C**). ^##^ indicates statistically significant difference compared with normal group at *p* < 0.01. ** represents statistically significant difference compared with OVA group at *p* < 0.01.

**Figure 2 foods-14-00016-f002:**
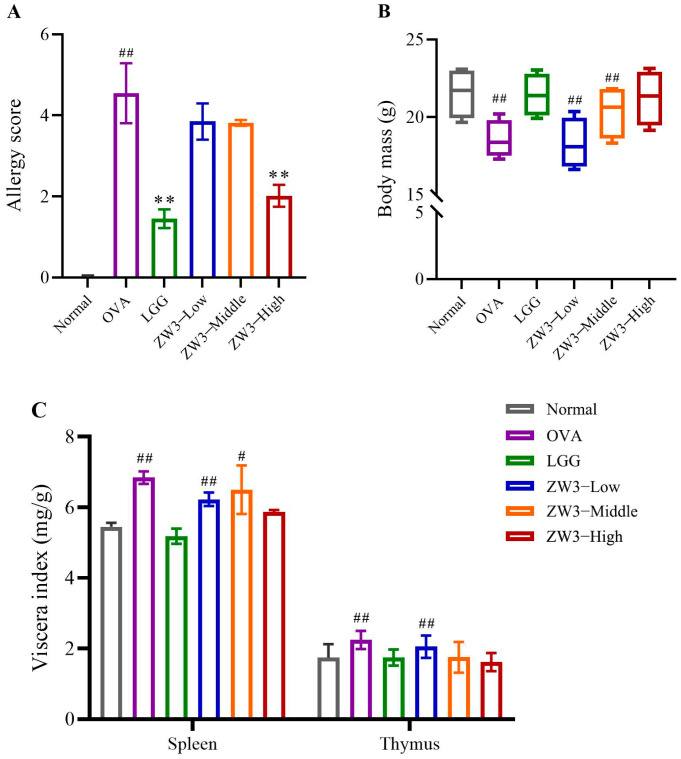
The allergic symptoms (**A**), body mass (**B**), and spleen and thymus indexes (**C**) in all groups. ^#^ and ^##^ indicate statistically significant differences compared with control group at *p* < 0.05 and *p* < 0.01, respectively. ** represents statistically significant difference compared with OVA group at *p* < 0.01.

**Figure 3 foods-14-00016-f003:**
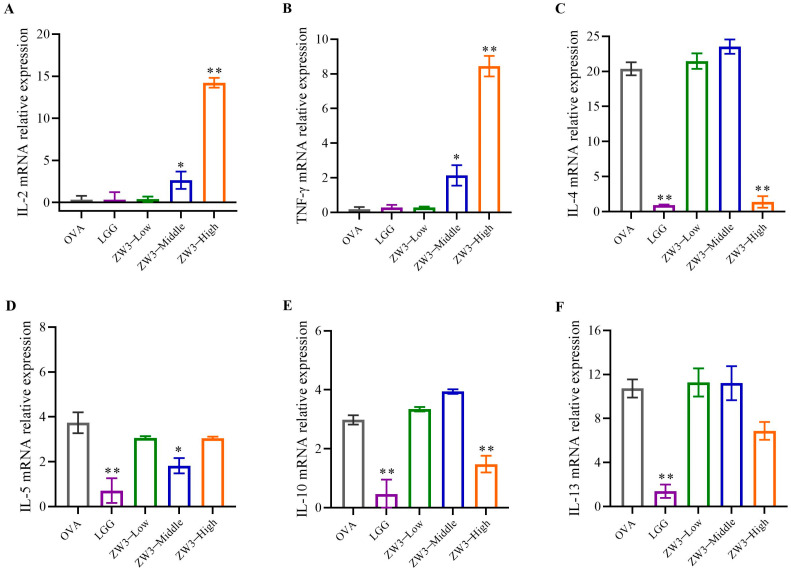
Relative mRNA expression of IL-2 (**A**), IFN-γ (**B**), IL-4 (**C**), IL-5 (**D**), IL-10 (**E**), and IL-13 (**F**) in spleen in treated and OVA mice. * and ** indicate statistically significant compared with OVA group at *p* < 0.05 and *p* < 0.01, respectively.

**Figure 4 foods-14-00016-f004:**
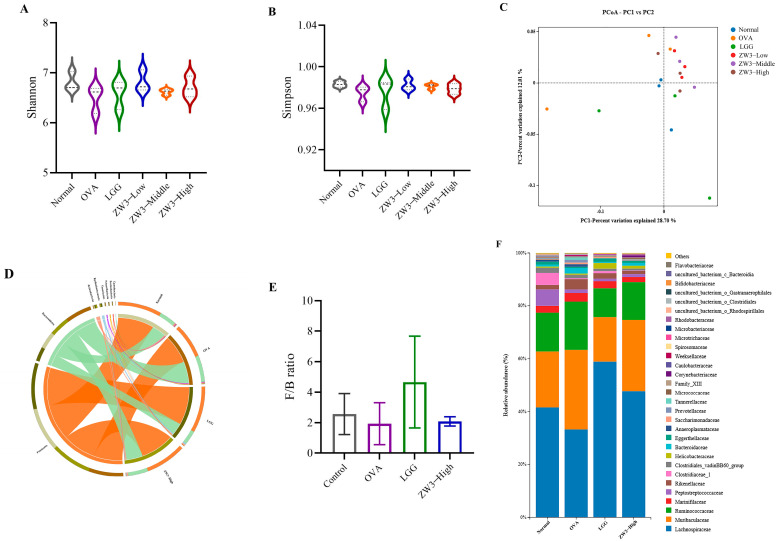
ZW3 modulates intestinal flora in OVA-sensitized mice. Shannon index (**A**) and Simpson index (**B**), reflecting the α-diversity of intestinal flora. (**C**) PCoA analysis reflecting the β-diversity of intestinal flora. (**D**) Bacterial composition at the phylum level. (**E**) B/F radio. (**F**) Bacterial composition at the family level.

**Table 1 foods-14-00016-t001:** Gene names, primers, and amplification characteristics of RT-qPCR.

Target Gene	Primer Sequence (5′-3′)	Tm (°C)
IL-2	F: TCCACTTCAAGCTCTACAG R: GAGTCAAAGTCCAGAACATGCC	60
IFN-γ	F: CTGGCAGGATGATTCTGCTGG R: GCATACGACAGGGTTCAAGTTAT	60
IL-4	F: ACAGGAGAAGGGACGCCAT R: GAAGCCCTACAGACGAGCTCA	60
IL-5	F: AAGGATGCTTCTGCACTTGA R: TTACTCTGCTACTCCGAAGG	60
IL-10	F: AGGGCACCCAGTCTGAGAACA R: CGGCCTTGCTCTTGTTTTCAC	60
IL-13	F: AGCATGGTATGGAGTGTGGA R: TGGTTGTAGAGGTTAACGTT	60
GAPDH	F: ATGGTGAAGGTCGGTGTGAACG R: CGCTCCTGGAAGATGGTGATGG	60

## Data Availability

The original contributions presented in this study are included in the article. Further inquiries can be directed to the corresponding author.

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
