# Peer review of "Amelioration Effect of Lactobacillus kefiranofaciens ZW3 on Ovalbumin-Induced Allergic Symptoms in BALB/c Mice"

_foods, 2024, doi:10.3390/foods14010016_

Round 1
Reviewer 1 Report
Comments and Suggestions for Authors
The article shows similarity to that of Hong et al., 2010 (DOI: 10.1111/j.1750-3841.2010.01787.x). However, this study employs a different strain (Lactobacillus kefiranofaciens ZW3), which I believe gives it significant potential. Furthermore, I consider one of its strengths to be the inclusion of microbiota analyses. However, I believe the manuscript should be supplemented to explicitly acknowledge similar previous studies, highlighting how it differs and what novel insights it contributes to the field.
Below, I propose the following recommendations and corrections by section:
Abstract.
In line 11 the meaning of LLG is not specified
In line 15 “spleen indexs” should be corrected to “spleen indexes”.
In nine 103 “control” should be corrected to “control”.
Introduction
I believe the introduction could be improved by providing a greater focus on how Lactobacillus kefiranofaciensZW3 differs from other previously studied strains. For example, there is no reference to the work of Hong et al., 2010 (DOI: 10.1111/j.1750-3841.2010.01787.x), despite the studies being highly similar.
Methods
The research design is appropriate for the study's objective, as it uses BALB/c mice, a well-established model for food allergy research. The study employs proper controls and treatments with varying doses to evaluate dose-dependent effects.
However, it is not specified whether normality tests were conducted on the data. This is a critical aspect that should be addressed to ensure that the data were analyzed appropriately.
Results
In line 214 “in” preposition should be corrected to “on”.
Discussion
In line 240 “enhanced spleen and thymus indexs” should be corrected to “enhanced spleen and thymus indexes”
in line 252 “active basophils and mast cells” should be corrected to “activates basophils and mast cells”.
I believe the discussion should delve deeper into how the results or findings could have potential applications, whether in the clinical field or the food industry.
Conclusion
It would be worthwhile to highlighting the strengths of the study as well as its limitations I rode to contextualize the results.
In line 283 “increase” and “decrease” should be corrected to “increases” and “decreases”.
Author Response
Dear Reviewer:
We greatly appreciate the efforts you have done for our manuscript foods-3317548 “Amelioration effect of Lactobacillus kefiranofaciens ZW3 on ovalbumin induced allergic symptoms in Balb/c mice”.
We are truly grateful to you for your critical comments and valuable suggestions. Based on these comments and suggestions, we have made improvements of the manuscript as follows and all changes are highlighted in yellow in the revised manuscript. In addition, we decrease the repetitive rate of manuscript to below 30%, and the repetition rate of a single article is less than 5%, and checked it with the turnitin database, all changes are highlighted in blue in the revised manuscript. We hope the new manuscript will meet your magazine’s standard.
We are looking forward to hearing from you regarding our submission. We would glad to respond to any further questions and comments you may have.
The point-by-point answers are shown as follows:
Abstract.
Point 1: In line 11 the meaning of LLG is not specified.
Response 1: Thank you for your suggestion. In this study, LGG refers to the strain Lactobacillus rhamnosus GG, which was widely known for its anti-allergic activity [1-3]. The full name and detailed information of LGG was included in the text (Page 1 line 11-12).
Point 2: In line 15 “spleen indexs” should be corrected to “spleen indexes”.
Response 2: Thanks for your kind suggestion. We have corrected the mistakes and checked the entire manuscript. Please see Page 1 line 16; Page 3 line 96; 103; Page 5 line 163; 172; Page 6 line 180; 183; Page 8 line 243 in the text.
Point 3: In nine 103 “control” should be corrected to “control”.
Response 3: Thanks for your suggestion. It has been revised. Please see Page 1 line 13 in the text.
Introduction
Point 4: I believe the introduction could be improved by providing a greater focus on how Lactobacillus kefiranofaciens ZW3 differs from other previously studied strains. For example, there is no reference to the work of Hong et al., 2010 (DOI: 10.1111/j.1750-3841.2010.01787.x), despite the studies being highly similar.
Response 4: Thank you for your helpful suggestion. The “Introduction” and “Discussion” sections have been revised and the Hong et al., 2010 (DOI: 10.1111/j.1750-3841.2010.01787.x) reference was added.
Firstly, We cite this reference in “Introduction” section to demonstrate the anti-allergic potential of the probiotic Lactobacillus kefiranofaciens. The strain ZW3 is another subspecies (kefiranofaciens) of Lactobacillus kefiranofaciens different from Lactobacillus kefiranofaciens M1 (kefirgranum) (Page 2 line 49-54).
Secondly, We cite this reference in “Discussion” section to show that we achieved similar results with this study, but we further analysed the possibility that ZW3 may exert its anti-allergic activity by targeting the intestinal flora (Page 10 line 289-302).
Finally, we further described in the “Introduction” section that ZW3, unlike other strains, has good ability in intestinal colonisation and regulating intestinal flora and has great potential in modulating body immunity, improving depression, non-alcoholic fatty liver and many other diseases, which is why we chose this strain (Page 2 line 55-63).
Methods
Point 5: The research design is appropriate for the study's objective, as it uses BALB/c mice, a well-established model for food allergy research. The study employs proper controls and treatments with varying doses to evaluate dose-dependent effects.
However, it is not specified whether normality tests were conducted on the data. This is a critical aspect that should be addressed to ensure that the data were analyzed appropriately.
Response 5: Thanks for your valuable suggestion. In data analysis, we used SPSS 23.0 software to test for normality. The relevant methods have been supplemented in section “Statistical Analysis” (Page 4 line 128-130).
Results
Point 6: In line 214 “in” preposition should be corrected to “on”.
Response 6: Thanks for your suggestion. It has been revised (Page 7 line 211).
Discussion
Point 7: In line 240 “enhanced spleen and thymus indexs” should be corrected to “enhanced spleen and thymus indexes”
Response 7: Thanks for your kind suggestion. They have been revised (Page 8 line 243).
Point 8: In line 252 “active basophils and mast cells” should be corrected to “activates basophils and mast cells”.
Response 8: Thanks for your useful suggestion. We have corrected this mistake (Page 9 line 255).
Point 9: I believe the discussion should delve deeper into how the results or findings could have potential applications, whether in the clinical field or the food industry.
Response 9: Thanks for your professional suggestion. We revised the “Discussion” section, further discussed the potential applications of our research in the food industry (Page 10 line 281-302). This study proved that the activity of L. kefiranofaciens in preventing and treating allergy symptoms may be related to the ability to regulate intestinal flora. These results will help to provide new insights into the effect of L. kefiranofaciens on improvement food allergies and provide better guidance for the development of L. kefiranofaciens probiotic products.
Conclusion
Point 10: It would be worthwhile to highlighting the strengths of the study as well as its limitations I rode to contextualize the results.
Response 10: Thanks for your helpful suggestion. We have added some descriptions in the “Results” section that focus on the strengths and limitations of our study (Page 10 line 312-316). Firstly, our study demonstrated that the anti-allergic effect of L. kefiranofaciens may be related to the intestinal flora, which is our strength. Secondly, the conclusions obtained in this paper still require further clinical validation, which is our limitation.
Point 11: In line 283 “increase” and “decrease” should be corrected to “increases” and “decreases”.
Response 11: Thanks for your kind suggestion. We have corrected this mistake (Page 10 line 309-310).
Finally, we sincerely appreciate your positive evaluations and insightful comments. The comments are all valuable and very helpful in revising and improving our manuscript, as well as providing important guidance in our research.
References:
[1] CHEN X X, ZHAO X L, HU Y Z, et al. Lactobacillus rhamnosus GG alleviates β-conglycinin-induced allergy by regulating the T cell receptor signaling pathway [J]. Food & Function, 2020, 11(12): 10554-67.
[2] NOCERINO R, COPPOLA S, CARUCCI L, et al. The Step-down approach in children with cow's milk allergy: Results of a randomized controlled trial [J]. Allergy, 2023.
[3] TAN W F, ZHOU Z C, LI W, et al. Lactobacillus rhamnosus GG for Cow's Milk Allergy in Children: A Systematic Review and Meta-Analysis [J]. Frontiers in Pediatrics, 2021, 9.

Reviewer 2 Report
Comments and Suggestions for Authors
i would like to see one more control..a group of healthy mice receiving ZW3 without OVA sensitization would help confirm that ZW3’s effects are specific to allergic conditions.

Author Response
Dear Reviewer:
We greatly appreciate the efforts you have done for our manuscript foods-3317548 “Amelioration effect of Lactobacillus kefiranofaciens ZW3 on ovalbumin induced allergic symptoms in Balb/c mice”.
We are truly grateful to you for your critical comments and valuable suggestions. Based on these comments and suggestions, we have made improvements of the manuscript as follows and all changes are highlighted in yellow in the revised manuscript. In addition, we decrease the repetitive rate of manuscript to below 30%, and the repetition rate of a single article is less than 5%, and checked it with the turnitin database, all changes are highlighted in blue in the revised manuscript. We hope the new manuscript will meet your magazine’s standard.
We are looking forward to hearing from you regarding our submission. We would glad to respond to any further questions and comments you may have.
The point-by-point answers are shown as follows:
Abstract
Point 1: Please check font in text.
Response 1: Thanks for your helpful suggestion. We have corrected the mistakes and checked the entire manuscript.
Introduction
Point 2: In line 37, The sentence is repeated see line 30.
In line 39, I don’t understand what authors wanted to say here.
In line 44, I understand what you wanted to say but please make the flow better.
In line 53, please rewrite this whole paragraph.
Response 2: Thank you very much for your valuable comment. The whole paragraph which included line 37, line 39, line 44 and line 53, has been rewritten (Page 2 line 41-54). In the revised paragraph, we firstly introduced that the imbalance of the intestinal flora may lead to allergic reactions. Then, the widely recognized effects of probiotics on targeting the intestinal flora was introduced. Lastly, we highlighted that probiotics may improve allergic symptoms by regulating intestinal flora.
Methods
Point 3: In line 70, ‘cfu’ large letter.
Response 3: Thanks for your useful suggestion. We corrected the mistakes and checked the entire manuscript (Page 2 line 73).
Point 4: In line 73, why only females, why not males.
Response 4: We sincerely appreciate your insightful question. Firstly, the reason we chose female mice was because our method of establishing allergic mouse model was referred to the experimental approach of Orgel et al. (2019)[1] and Guo et al. (2023)[2]. And we added these references in manuscript (Page 3 line 89). Secondly, it has been shown that females suffered more frequently from allergic symptoms than males in human and mice due to hormonal effects, gender-specific behaviour, and so on[3, 4].
Point 5: In line 75, ‘12-h/12-hlight/dark cycle’ space.
Response 5: Thanks for your kind suggestion. We have corrected this mistake (Page 2 line 77).
Point 6: In line 82, ‘200 μL/d’ should be corrected to ‘200 μL/day’.
Response 6: Thanks for your useful suggestion. We have corrected this mistake and checked the entire manuscript (Page 3 line 84; 92).
Point 7: In line 104, ‘10000 r/min’ should be corrected to ‘10000 rpm/min’.
Response 7: Thanks for your kind suggestion. We have corrected this mistake and checked the entire manuscript (Page 3 line 107; 110).
Point 8: In line 105, ‘(Future, China)’ add city.
Response 8: Thanks for your valuable suggestion. We have added the description of the city of reagent company in the manuscript and checked the entire manuscript (Page 3 line 108; 110; 114; 119).
Results
Point 9: In line 130, I would like to see one more control, a group of healthy mice receiving ZW3 without OVA sensitization would help confirm that ZW3’s effects are specific to allergic conditions.
Response 9: Thanks for your professional suggestion.
Firstly, the main purpose of this study was to investigate whether ZW3 could treat food allergy, so there was no prevention group, and the experimental method was referred to Guo et al. (2023)[2]. However, this is also a limitation of our study. We should subsequently design rational animal experiments and clinical trials to explore whether ZW3 can prevent the development of allergy.
Secondly, previous studies have shown that intestinal dysbiosis generally precedes the allergic reaction, so in this paper we wanted to investigate whether ZW3 could reverse this dysbiosis to improve allergy symptoms. However, this dysbiosis is not present in healthy mice.
Thirdly, Our previous studies have demonstrated that ZW3 has intestinal colonisation and improved intestinal flora in BALB/C healthy mice, which is why we chose this strain[5].
Finally, we suggested that the anti-allergic effect of ZW3 may not be specific. The alleviation of allergic symptoms by ZW3 may be due to its overall modulating effect on the body, such as improving the intestinal flora and modulating the body's immunity, among other activities.
Point 10: In line 144, Please rephrase ‘The mice were executed on the 22nd day, and the serum was collected for the detection of immunologic factors by ELISA to monitor the effects of ZW3 administration’.
Response 10: Thanks for your valuable suggestion. We have rephrased this sentence (Page 5 line 146-147).
Finally, we sincerely appreciate your positive evaluations and insightful comments. The comments are all valuable and very helpful in revising and improving our manuscript, as well as providing important guidance in our research.
References:
[1] ORGEL K, SMEEKENS J M, YE P, et al. Genetic diversity between mouse strains allows identification of the CC027/GeniUnc strain as an orally reactive model of peanut allergy [J]. Journal of Allergy and Clinical Immunology, 2019, 143(3): 1027-+.
[2] GUO Z H, WANG Q, ZHAO J H, et al. Lactic acid bacteria with probiotic characteristics in fermented dairy products reduce cow milk allergy [J]. Food Bioscience, 2023, 55.
[3] MELGERT B N, POSTMA D S, KUIPERS I, et al. Female mice are more susceptible to the development of allergic airway inflammation than male mice [J]. Clinical and Experimental Allergy, 2005, 35(11): 1496-503.
[4] AFIFY S M, PALI-SCHöLL I. Adverse reactions to food: the female dominance - A secondary publication and update [J]. World Allergy Organization Journal, 2017, 10.
[5] XING Z Q, TANG W, YANG Y, et al. Colonization and Gut Flora Modulation of Lactobacillus kefiranofaciens ZW3 in the Intestinal Tract of Mice [J]. Probiotics and Antimicrobial Proteins, 2018, 10(2): 374-82.

Round 2
Reviewer 1 Report
Comments and Suggestions for Authors
I have reviewed the responses provided and observed that most of the suggestions have been addressed. However, it remains unclear which normality test was applied; the response only specifies the software used for the analysis (SPSS in this case). Please clarify this point to ensure a complete and transparent methodology.
Author Response
Dear Reviewer:
We greatly appreciate the efforts you have done for our manuscript foods-3317548 “Amelioration effect of Lactobacillus kefiranofaciens ZW3 on ovalbumin induced allergic symptoms in Balb/c mice”.
We are truly grateful to you for your critical comments and valuable suggestions. Based on the suggestion that ‘manuscript remains unclear which normality test was applied’, we have made improvements of the manuscript as follow: we clearly describe that Shapiro-Wilk test was performed to normality test and all changes are highlighted in yellow in page 5 line 154-155 of revised manuscript. We hope the new manuscript will meet your magazine’s standard.
We are looking forward to hearing from you regarding our submission. We would glad to respond to any further questions and comments you may have.
With best regards,
Yours sincerely,
Dr Weitao Geng
